# Cerebral Biomarkers and Blood-Brain Barrier Integrity in Preeclampsia

**DOI:** 10.3390/cells11050789

**Published:** 2022-02-24

**Authors:** Therese Friis, Anna-Karin Wikström, Jesenia Acurio, José León, Henrik Zetterberg, Kaj Blennow, Maria Nelander, Helena Åkerud, Helena Kaihola, Catherine Cluver, Felipe Troncoso, Pablo Torres-Vergara, Carlos Escudero, Lina Bergman

**Affiliations:** 1Department of Women’s and Children’s Health, Uppsala University, 75185 Uppsala, Sweden; anna-karin.wikstrom@kbh.uu.se (A.-K.W.); maria.nelander@kbh.uu.se (M.N.); lina.bergman@kbh.uu.se (L.B.); 2Vascular Physiology Laboratory, Department of Basic Sciences, Faculty of Sciences, University of Bío-Bío, Chillán 3810178, Chile; jeseniacurio2@gmail.com (J.A.); joseman1115@gmail.com (J.L.); fetronc@gmail.com (F.T.); cescudero@ubiobio.cl (C.E.); 3Group of Research and Innovation in Vascular Health (GRIVAS Health), Chillán 3810178, Chile; pabltorr@udec.cl; 4Escuela de Enfermería, Facultad de Salud, Universidad Santo Tomás, Los Ángeles 4441171, Chile; 5Department of Psychiatry and Neurochemistry, Institute of Neuroscience and Physiology, The Sahlgrenska Academy at the University of Gothenburg, 43180 Mölndal, Sweden; henrik.zetterberg@clinchem.gu.se (H.Z.); kaj.blennow@neuro.gu.se (K.B.); 6Clinical Neurochemistry Laboratory, Sahlgrenska University Hospital, 43180 Mölndal, Sweden; 7Department of Neurodegenerative Disease, UCL Queen Square Institute of Neurology, Queen Square, London WC1E 6BT, UK; 8UK Dementia Research Institute at UCL, London WC1E 6BT, UK; 9Hong Kong Center for Neurodegenerative Diseases, Clear Water Bay, Hong Kong, China; 10Department of Immunology, Genetics and Pathology, Uppsala University, 75185 Uppsala, Sweden; helena.akerud@gynhalsan.se (H.Å.); helena.kaihola@outlook.com (H.K.); 11Department of Obstetrics and Gynecology, Stellenbosch University, Cape Town 7500, South Africa; cathycluver@hotmail.com; 12Department of Pharmacy, Faculty of Pharmacy, University of Concepción, Concepción 4070386, Chile; 13Department of Obstetrics and Gynecology, Gothenburg University, 41650 Gothenburg, Sweden

**Keywords:** blood-brain barrier, preeclampsia, pregnancy, in vitro studies, cerebral biomarkers, NfL, tau, NSE, S100B

## Abstract

Cerebral complications in preeclampsia contribute substantially to maternal mortality and morbidity. There is a lack of reliable and accessible predictors for preeclampsia-related cerebral complications. In this study, plasma from women with preeclampsia (n = 28), women with normal pregnancies (n = 28) and non-pregnant women (n = 16) was analyzed for concentrations of the cerebral biomarkers neurofilament light (NfL), tau, neuron-specific enolase (NSE) and S100B. Then, an in vitro blood–brain barrier (BBB) model, based on the human cerebral microvascular endothelial cell line (hCMEC/D3), was employed to assess the effect of plasma from the three study groups. Transendothelial electrical resistance (TEER) was used as an estimation of BBB integrity. NfL and tau are proteins expressed in axons, NSE in neurons and S100B in glial cells and are used as biomarkers for neurological injury in other diseases such as dementia, traumatic brain injury and hypoxic brain injury. Plasma concentrations of NfL, tau, NSE and S100B were all higher in women with preeclampsia compared with women with normal pregnancies (8.85 vs. 5.25 ng/L, *p* < 0.001; 2.90 vs. 2.40 ng/L, *p* < 0.05; 3.50 vs. 2.37 µg/L, *p* < 0.001 and 0.08 vs. 0.05 µg/L, *p* < 0.01, respectively). Plasma concentrations of NfL were also higher in women with preeclampsia compared with non-pregnant women (*p* < 0.001). Higher plasma concentrations of the cerebral biomarker NfL were associated with decreased TEER (*p* = 0.002) in an in vitro model of the BBB, a finding which indicates that NfL could be a promising biomarker for BBB alterations in preeclampsia.

## 1. Introduction

Preeclampsia affects 3–5% of all pregnancies and is one of the most common causes of maternal and perinatal morbidity and mortality [1]. Preeclampsia is defined as de novo hypertension after 20 weeks of gestation accompanied by signs of maternal organ dysfunction, such as renal insufficiency, liver dysfunction, neurological features, hematological complications or fetal growth restriction as a sign of uteroplacental dysfunction [2,3]. Annually, more than 70,000 maternal deaths are associated with hypertensive disorders of pregnancy, where cerebral complications due to preeclampsia, such as eclampsia, cerebral edema and cerebral hemorrhage are leading causes of maternal death [3,4,5]. In the majority of cases, eclampsia is preceded by hypertension and in some cases by neurological symptoms; however, eclampsia can occur before the onset of hypertension and often in the absence of premonitory symptoms [6]. There is a lack of objective biomarkers for cerebral complications to preeclampsia and the underlying pathophysiology remains partly unknown.

The cerebral biomarkers neurofilament light (NfL), tau, neuron-specific enolase (NSE) and S100B are all present within cells of the central nervous system, the last of these in the astrocytic endfeet contributing to the neurovascular unit of the blood–brain barrier (BBB) [7]. They have all been extensively studied in the context of hypoxic or traumatic brain injury, different types of dementia and epilepsy. NfL and tau are both axonal proteins used as biomarkers for neurodegenerative disease [8,9]. In patients with traumatic brain injury, increased concentrations of NfL have been detected in both cerebrospinal fluid (CSF) and peripheral blood, compared with controls [10]. NSE is a glycolytic enzyme mainly found in neurons [11], which has proved useful in prognostication of patients with cardiac arrest and hypoxic ischemic encephalopathy [12]. Similarly, tau has been identified to predict six-month outcomes regarding cerebral symptoms after cardiac arrest [13]. S100B and NSE have both been used as predictors for poor neurological outcome after traumatic brain injury [14]. S100B might enter the circulation after an isolated BBB injury with loss of BBB integrity, even without injury to the brain parenchyma [15].

Previous studies exploring cerebral biomarkers for preeclampsia have reported increased plasma concentrations before the onset of disease [16,17,18], during disease [19,20] and one year postpartum [21] in women with preeclampsia compared with women with normal pregnancies. Studies have also reported higher S100B plasma concentrations in women with severe preeclampsia and eclampsia compared with women with preeclampsia without severe features [20,22]. In addition, neurological symptoms, such as visual disturbances, have been correlated with increased S100B plasma concentrations [19,23].

The BBB is a highly restrictive and specialized neurovascular network that isolates and protects the brain parenchyma from potential harmful molecules present in the systemic circulation. Disruption of the BBB in preeclampsia and eclampsia has been studied in both laboratory and clinical settings, mainly with animal models [24], or ex vivo animal models exposed to plasma from women with preeclampsia. Cerebral edema, commonly seen in eclampsia and sometimes in severe preeclampsia, may partly be due to disruption of the BBB, resulting in increased BBB permeability and passage of fluid into the brain parenchyma [25,26]. Up until recently there has been a paucity of studies assessing the human BBB in preeclampsia, due to difficulties in studying the BBB in a clinical setting. However, our research group has recently presented promising results on the human cerebral microvascular endothelial cell line (hCMEC/D3) [27,28,29] as a new in vitro model of the BBB in preeclampsia research [30,31].

It is still not known if cerebral biomarkers are useful in reflecting BBB alterations in preeclampsia and to our knowledge, cerebral biomarkers have never before been correlated to any measures of BBB integrity in preeclampsia. Therefore, the aim of this study was to correlate plasma concentrations of cerebral biomarkers in women with preeclampsia, women with normal pregnancies and non-pregnant women with BBB integrity, measured as changes in transendothelial electrical resistance (TEER) in the hCMEC/D3 in vitro model.

## 2. Materials and Methods

### 2.1. Study Population

The study population consisted of pregnant women with preeclampsia (n = 28) diagnosed according to the International Society for Studies on Hypertension in Pregnancy (ISSHP) 2018 guidelines [3]. Diagnostic criteria were de novo hypertension (systolic blood pressure (SBP) > 140 and/or diastolic blood pressure (DBP) > 90 mmHg) in combination with significant proteinuria (protein level > 300 mg/24 h or urine dipstick > 1+) after 20 gestational weeks. Although clinical preeclampsia can be diagnosed in the absence of proteinuria, with other signs of maternal organ dysfunction present, it has been recommended that proteinuria is used for patients enrolled in scientific research to ensure more specificity around the diagnosis [2]. These were also the common criteria used for preeclampsia diagnosis in the obstetric clinic at the time when cases were enrolled in the study.

Severe preeclampsia was defined according to the guidelines from the International Society for the study of Hypertension in Pregnancy (ISSHP) [32]. Criteria were an SBP ≥ 160 and/or DBP ≥ 110, development of HELLP syndrome, eclampsia or other severe organ manifestations.

As controls, women with normal pregnancies (n = 28), matched for maternal age and gestational length at inclusion, and non-pregnant women (=16) were recruited. The definition of a normal pregnancy required that the woman remained normotensive throughout her pregnancy. The pregnancy also had to result in term delivery (gestational week ≥ 37) of an infant with normal birth weight (±2 standard deviations of the mean birth weight for gestational age and sex) [33]. Women with prior hypertensive disorder in pregnancy were not included in any of the control groups. Further, none of the study groups included women with chronic hypertension, diabetes mellitus or chronic kidney disease.

The women were all recruited from the obstetric ward or the outpatient clinic at Uppsala University Hospital, Sweden between 2013 and 2016 [34].

Uppsala Ethical Review Board approved the study and informed consent was obtained from all participants.

### 2.2. Sample Collection

Plasma samples were collected in Vacutainer tubes with lithium heparin (Becton, Dickinson, Franklin Lakes, NJ, USA) within four hours of study inclusion. The samples were centrifuged for 10 min at 1500 *g* and the plasma was immediately frozen at −70 °C for later analysis. Thawed plasma was used for analysis of the cerebral biomarkers. NfL and tau concentrations were measured in Mölndal, Sweden, whereas NSE and S100B concentrations were measured in Uppsala, Sweden. Further, frozen plasma was shipped to Chillán, Chile, thawed and added to the BBB model for TEER measurements.

### 2.3. Biomarker Assay

Plasma NfL concentration was measured with an in-house single molecule array (Simoa) method, whilst plasma tau concentration was measured with the Human Total Tau 2.0 kit and the Simoa platform (Quanterix, Billerica, MA, USA), both previously described in detail [8,35]. Laboratory technicians, who were blinded to clinical data, performed measurements in one round of experiments, using one batch of reagents. Two quality-control samples were run in duplicates in the beginning and the end of each run, showing coefficients of variance (CVs) for intermediate precision of 6.0% at 8.5 pg/mL and 5.1% at 121 pg/mL for NfL, whereas CVs were 7.3% at 32.2 pg/mL and 7.0% at 7.5 pg/mL for tau.

Plasma NSE and S100B concentrations were measured by an enzyme-linked immunosorbent assay (ELISA). A commercially available kit (Sangtec 100 Elisa, Diasorin, MN, USA) was used and the samples were run according to the manufacturer’s recommendation. The intra- and inter-assay coefficients of variation were 2.8% and 4.3% and 4.6% and 3.1%, respectively, for NSE and S100B.

### 2.4. hCMEC/D3 In Vitro Model

The hCMEC/D3 cell line (Merck Millipore, Darmstadt, Germany) [27] was used for the in vitro experiments. Monolayers of cells were seeded on semipermeable plates coated with rat-tail type I collagen (Discovery Labware, Bedford, MA, USA) at a density of 20,000 cells/well. Medium EndoGro MV Supplement Kit (Merck Millipore) was used as a culturing medium, and cells were incubated at 37 °C, 5% CO_2_. Once cells reached 100% of confluence, and a TEER value larger than 20 Ωcm^2^, they were used for experiments. In addition, six hours prior to experiments the culture medium was replaced by a medium without growth supplements. Cultured hCMEC/D3 cells were treated (1%, *v*/*v*, 12 h) with thawed plasma either from women with preeclampsia, women with normal pregnancies or non-pregnant women. An epithelial Volt/Ohm meter (EVOM2, World Precision Instruments, Sarasota, FL, USA) with two electrodes in each compartment was used to measure the TEER. Measurements were performed both before adding plasma (baseline), and after incubation with plasma. Delta-TEER values were calculated by subtracting basal TEER values from TEER values after exposure to plasma.

Confirmatory experiments of TEER and cell permeability to high-molecular weight fluorescent dye (Fluorescein-5-isothiocyanate FITC-dextran 70 kDa) were performed as previously reported [30] using randomly selected plasmas from women with preeclampsia (n = 12), and women with normal pregnancies (n = 13).

The hCMEC/D3 cell line was used in passages 5 to 10. Individual plasmas were used in duplicate experimental replicates. None of the plasmas were excluded at the final analysis. More detailed experimental conditions are described in a previous publication [30].

### 2.5. Statistical Analyses

Background characteristics were presented as medians with interquartile range (IQR) and numbers with percentage (%) as appropriate. Groups were compared by one-way ANOVA, Chi-Square or Kruskal–Wallis test as appropriate, and the significance level was set at 0.05.

Plasma concentrations of the cerebral biomarkers (NfL, tau, NSE and S100B), TEER and permeability were presented as medians with interquartile range (IQR). Differences between groups were compared by non-parametric analysis by Kruskal–Wallis test and pairwise comparisons by Mann-Whitney U-test. In case of statistical significance, a Bonferroni post-hoc test was used.

Associations between concentrations of the cerebral biomarkers NfL, tau, NSE and S100B, and TEER values were analyzed with a cumulative probability model [36] allowing for different, possibly non-linear, associations in the three groups of women. The model was further adjusted for baseline TEER, and for the confounders maternal age, parity and BMI. The confounders were identified with a DAG (directed acyclic graph). For this fully flexible model a *p*-value for all non-linear terms of 0.835 was calculated. The model was subsequently adjusted to a model where all associations were linear. Data and statistical analyses were performed with IBM SPSS Statistics for Windows, Version 25.0 (IBM Corp, Armonk, NY, USA), GraphPad Prism 6.00 (GraphPad Software, San Diego, CA, USA) and R version 3.6.1 with the add-on package rms [37,38].

Subgroup analyses of the women with preeclampsia were performed with regards to clinical symptoms in association with cerebral biomarkers and change in TEER, i.e., the difference between measured TEER before and after adding plasma to the model. Analyses were performed with the Mann-Whitney U-test.

## 3. Results

### 3.1. Background Characteristics

Clinical characteristics of the study population are described in Table 1 and Table 2. There was no difference in maternal age or gestational length at inclusion between the pregnant study groups. Women with preeclampsia had a higher early-pregnancy body mass index (BMI) and were more often nulliparous compared with women with normal pregnancies and non-pregnant women. In the group of women with preeclampsia most women had antihypertensive treatment (79%), and at inclusion approximately one third of them had severe headaches and/or visual disturbances. At delivery, 57% of the women had developed severe preeclampsia, with a recorded SBP ≥160 and/or DBP ≥110 [32]. None of the women developed eclampsia, neurological complications or other severe organ manifestations, and none required magnesium sulphate prophylaxis against imminent eclampsia.

### 3.2. Plasma Concentrations of NfL, Tau, NSE and S100B

Compared with women with normal pregnancies, plasma concentrations of the cerebral biomarkers were higher in plasma from women with preeclampsia; NfL (5.25 ng/L, IQR 3.93–7.63 ng/L vs. 8.85 ng/L, IQR 6.78–12.65 ng/L, *p* < 0.001); tau (2.40 ng/L, IQR 1.80–2.58 ng/L vs. 2.90 ng/L, IQR 2.40–4.35 ng/L, *p* < 0.05); NSE (2.37 µg/L, IQR 1.93–2.85 µg/L vs. 3.50 µg/L, IQR 2.84–4.55 µg/L, *p* < 0.001) and S100B (0.05 µg/L, IQR 0.03–0.08 µg/L vs. 0.08 µg/L, IQR 0.05–0.10 µg/L, *p* < 0.01) (Figure 1a–d). In addition, plasma concentrations of NfL were also higher in women with preeclampsia compared with non-pregnant women (8.85 ng/L, IQR 6.78–12.65 ng/L vs. 5.65 ng/L, IQR 4.83–6.40 ng/L, *p* < 0.001) (Figure 1a).

### 3.3. Effects of Plasma on TEER in an In Vitro BBB Model

The effects of plasma on changes in TEER, measured in the hCMEC/D3 in vitro model, in this population have been published previously [30]. Baseline TEER (i.e., TEER prior to exposure to plasma) was not statistically different among cell monolayers. After exposure to plasma, a significantly larger reduction in TEER was detected in cell monolayers exposed to plasma from women with preeclampsia compared with plasma from women with normal pregnancies and from non-pregnant women [30].

For this manuscript a re-confirmation analysis of previously published data was performed, where plasma was randomly included from the group of women with preeclampsia (PE, n = 12), and the women with normal pregnancies (NP, n = 13). Results are shown in Figure 2.

### 3.4. Association of NfL, Tau, NSE and S100B with TEER

The associations between cerebral biomarkers and TEER are presented in Figure 3. NfL was the only biomarker that showed a negative linear association with TEER, i.e., higher plasma concentrations of NfL correlated with a larger reduction in TEER (*p* < 0.01) after exposure to plasma. Data did not support associations between tau, NSE or S100B and TEER.

### 3.5. Subgroup Analyses of Neurological Symptoms in the Women with Preeclampsia

In the group of women with preeclampsia, 10 out of 28 women reported severe headaches (VAS ≥ 5), and 10 women reported visual disturbances.

Higher plasma concentrations of NfL were found in the women with preeclampsia that expressed severe headache than in the women with mild or no headache (11.65 ng/L, IQR 9.58–15.63 ng/L vs. 7.40 ng/L, IQR 5.93–9.93 ng/L, *p* = 0.024). The plasma concentrations of the other biomarkers (tau, NSE and S100B) did not relate to severity of headache. No associations were found between reported visual disturbances and any of the biomarkers. When comparing the women who had a combination of severe headache and visual disturbances with women who did not have this combination of symptoms, there were no differences in plasma concentrations of the respective biomarkers.

Women with preeclampsia that expressed severe headache (VAS ≥ 5) exhibited a greater reduction of TEER than the women with mild or no headache (10.17, IQR 7.32–10.45 vs. 7.74, IQR 5.65–9.42, *p* = 0.040). No associations were found between reported visual disturbances and change in TEER. However, a greater reduction of TEER was seen in the women who had a combination of severe headache and visual disturbances, compared with the women without this combination of symptoms (10.32, IQR 7.82–11.72 vs. 7.81, IQR 5.77–9.56, *p* < 0.021).

## 4. Discussion

### 4.1. Main Findings

In this study we demonstrated that circulating concentrations of NfL was associated with decreased TEER. In addition, we corroborated previous findings of higher plasma concentrations of the cerebral biomarkers NfL, tau, NSE and S100B in women with preeclampsia compared with women with normal pregnancies. Concentrations of NfL were also higher in women with preeclampsia compared with non-pregnant women. Further, women with preeclampsia that expressed severe headache had higher concentrations of plasma NfL and a greater reduction of TEER, compared with women with lesser symptoms. There was also a greater reduction of TEER in women with the combination of severe headache and visual disturbances.

### 4.2. Research Implications

Several previous studies have reported increased circulating concentrations of cerebral biomarkers in preeclampsia [16,17,18,20,22], findings that are also supported in this study. A novel finding, however, is that plasma concentrations of NfL were also significantly higher in plasma in women with preeclampsia compared with non-pregnant women. We did not find a difference in plasma concentrations of any of the biomarkers comparing women with normal pregnancies with non-pregnant women. Hence, pregnancy alone does not seem to have a major impact on circulating levels of these cerebral biomarkers.

NfL, tau, NSE and S100B have repeatedly been reported useful as circulating biomarkers of cerebral injury in other neurological disorders such as neurodegenerative disease, traumatic brain injury and epilepsy [8,9,10,14,15,39]. Previous studies of these biomarkers in relation to BBB injury have, however, most often evaluated biomarker concentrations in male or mixed populations with other neurological conditions than preeclampsia. The results from these studies seem to be somewhat conflicting regarding whether they support a BBB injury or not [40,41]. Our findings in a female population are suggestive of BBB alterations in terms of loss of ionic tightness, and possibly neuroaxonal injury, reflected by higher plasma concentrations of NfL in women with preeclampsia. We propose this by an association between higher circulating NfL concentrations in plasma and decreased TEER in an in vitro model of the BBB. However, this finding could have different underlying causes [15]. It could mean that intracerebral concentrations of NfL are normal, but are secreted over the BBB to a greater extent due to increased permeability, and are subsequently detected in higher concentrations in plasma. It could also reflect that NfL is released in higher concentrations in the central nervous system (CNS) following neuroaxonal injury, which might in turn cause the BBB to become more permeable or render increased NfL concentrations in plasma through alternative transport mehcanisms [15]. Alternatively, it may reflect a secondary breakdown of the BBB caused by circulating harmful molecules associated with preeclampsia, e.g., anti-angiogenic or inflammatory factors, including small extracellular vesicles [31], that allow NfL to cross from the brain to the circulation. It could also be caused by a combination of these proposed theories.

However, our findings could not support that higher plasma concentrations of NSE, S100B and tau had a significant association to decreased TEER. Thus, alternative explanations, such as other transport pathways across the BBB, dose-response-dependent passage, increased extracerebral existence or hemolysis in serum/plasma samples, have to be considered as causes of the higher concentrations of these biomarkers detected in women with preeclampsia. Furthermore, an in vitro model also has its own limitations and may not perfectly reflect in vivo conditions.

Recently published results from our research group reported increased concentrations of NfL in both plasma and CSF in women with preeclampsia compared with normotensive pregnant women. In addition, there was a strong correlation between concentrations of NfL in plasma and CSF [42]. Previous studies of women with preeclampsia have not been able to establish whether a BBB impairment exists in these women or not [43]; however, human studies are few on this subject.

Models for studying the brain and BBB function in preeclampsia that have found evidence of BBB injury have traditionally included in vitro models using animal endothelial cells, along with studies of brain hemodynamic regulation in humans and different preeclampsia-mimicking models in animals in vivo [24,25,26,29,44,45,46]. Recently our research group introduced the use of the hCMEC/D3 cell line as a means of studying the effect of preeclampsia on the BBB. We demonstrated a reduction of TEER, and increased permeability to 70 kDa FITC-dextran over the BBB, when a monolayer of cells was exposed to plasma from women with preeclampsia and compared with plasma from women with normal pregnancies and non-pregnant controls [30]. The expression of tight junction proteins (zonula occludens-1 (ZO-1) and occludin) and phosphorylation of two tyrosine residues of VEGFR2 (pY951 and pY1175) were also explored, with no changes in the mRNA expression of the tight junction proteins. However, changes in their localization morphology or function were not investigated [30].

For this study we re-confirmed findings of a reduction in TEER and increased permeability in cells exposed to plasma from women with preeclampsia (Figure 2). In addition, we examined expression of tight junction protein claudin-5. This was performed as a sole experiment with only four randomly chosen plasma samples. Brain endothelial cells exposed to plasma from women with preeclampsia demonstrated a reduced protein abundance of claudin-5 in the cell membrane, whereas it was enhanced in the cytoplasmic fraction (Appendix A).Characteristically, brain endothelial cells express higher levels of tight junction proteins than peripheral endothelial cells [47]. The key components of intercellular tight junctions are the transmembrane proteins occludin, claudin and junctional adhesion molecules (JAMs), which form complex strands that govern the permeability characteristics of the paracellular route [48,49]. Among them, claudins are directly responsible for the selective permeability of tight junctions [50,51]. There are 27 different claudin isoforms found in mammals [52]. Present in the cell membrane, claudin-5 increases TEER, primarily by decreasing the permeability of cations [52,53]. In order to maintain this restricted diffusion pathway, tight junctions are linked to the cytoplasmic zonula occludens proteins that provide a structural bridge to the actin cytoskeleton. Consequently, the degree of tightness is determined by the interactions of tight junction family members on endothelial cells.

Pre-clinical studies show conflicting results regarding the involvement of these tight junction proteins on the increased BBB permeability seen in preeclampsia models [54,55,56]. One study found that despite exhibiting increased water content and increased BBB permeability in the anterior cerebrum in a preeclampsia-like syndrome in rats, the protein expression of claudin-1, ZO-1 and occludin was not altered in that part of the brain. However, an upregulation of claudin-1 was detected in the posterior cerebrum of these animals, where brain water content was not altered [56].

In this manuscript, we report a reduction in the availability of claudin-5 in the cell membrane of brain endothelial cells exposed to plasma from women with preeclampsia. This may constitute an underlying mechanism for the reduction in TEER under these experimental conditions.

To our knowledge, no previous studies have explored the association between circulating concentrations of cerebral biomarkers and BBB alterations in women with preeclampsia. However, several studies have established a correlation between serum or plasma concentrations of S100B, which as of now is the most explored of the cerebral biomarkers in preeclampsia, and severity of disease and/or neurological symptoms [19,20,22,23].

### 4.3. Strengths and Limitations

Strengths in our study include the use of a human in vitro model of the BBB, as opposed to strictly animal-based models or the use of human plasma in animal models [24], to correlate BBB alterations in preeclampsia with peripheral biomarkers. This might reflect the human BBB to a higher extent compared with previous animal studies. Another strength is the well-characterized cohort of women and simultaneous measurements of cerebral biomarkers and BBB-assessment in the same woman.

A limitation of our study is that only one third of the women with preeclampsia in our cohort reported neurological symptoms, in the form of headaches or visual disturbances, and none developed severe cerebral complications [34]. The women in this study also underwent brain MRI scans, as part of previously published studies [34,57,58]. None of the women exhibited cerebral edema on their scans, which might reflect the fact that none of them suffered from severe neurological complications to preeclampsia, such as overt cerbral edema. Nevertheless, there were findings of decreased concentrations of magnesium (Mg) on magnetic spectroscopy and decreased perfusion in the caudate nucleus in the women with preeclampsia. Since the study population did not fully reflect the degree of CNS pathophysiology that can occur in preeclampsia, our results are only generalizable to preeclampsia without severe neurological engagement.

Unfortunately, there were no CSF samples available for this population. Measurements of NfL in CSF, and also the CSF/serum albumin ratio, in the same women, could possibly have helped address what the plasma NfL finding represents: neuroaxonal injury and/or leakage across the BBB.

For TEER measurements in this study we used a static in vitro model based on a monoculture. Future studies should include adaptations of the in vitro model to enhance the tightness of the BBB in order to better reflect in vivo conditions [27]. For example, co-culturing of hCMEC/D3 monolayers with astrocytes has been shown to increase TEER from 30 to 60 Ωcm^2^ [59], whereas exposure of hCMEC/D3 monolayers to a pulsatile flow in a capillary cartridge system resulted in TEER values of 1000–1200 Ωcm^2^ [60].

In this last regard, physical and technical parameters, such as setup, culture medium viscosity, electrodes and other instruments used, have been demonstrated to influence the measurement of TEER in BBB models. This may result in great variation of TEER results. Drawbacks of methods for TEER measurements have been further discussed by Vigh et al. [61]. To avoid these potential experimental pitfalls, all in vitro experiments were run in a blinded manner for the groups of plasma. In addition, analyses were performed head-to-head using plasmas from the three different groups, as previously reported [30].

### 4.4. Future Perspectives

Manifestations of acute cerebral complications of preeclampsia, such as eclampsia and cerebral hemorrhage or edema, are difficult to predict. A circulating biomarker, alone or in combination with other clinical data, that could identify women with preeclampsia at risk of such events, may in the future contribute to the prevention of severe cerebral complications. This would allow for a more customized management of women at high risk, e.g., by suggesting more aggressive treatment with antihypertensive drugs, administration of neuroprotective treatment such as MgSO_4_ [46], and/or aid in decisions about timing of delivery. Here we suggest that NfL could be a promising biomarker for BBB alterations and/or axonal injury in preeclampsia. However, it needs to be further explored in a population presenting with cerebral complications of the disease, and additional methods to assess BBB permeability and function should also be considered.

## 5. Conclusions

We demonstrated that plasma concentrations of NfL had a negative association with TEER in an in vitro model of the BBB. This finding supports the role of NfL as a promising biomarker to reflect cerebral involvement in women with preeclampsia. Further, in women with preeclampsia we found that those with severe headache, a possible sign of cerebral involvement, had both higher concentrations of plasma NfL and a greater reduction of TEER, compared with those without severe headache.

Prospective studies should be conducted to confirm the role of NfL as a biomarker for BBB alterations and/or neuroaxonal injury in women with preeclampsia, preferably in a population with manifest neurological complications.

## Figures and Tables

**Figure 1 cells-11-00789-f001:**
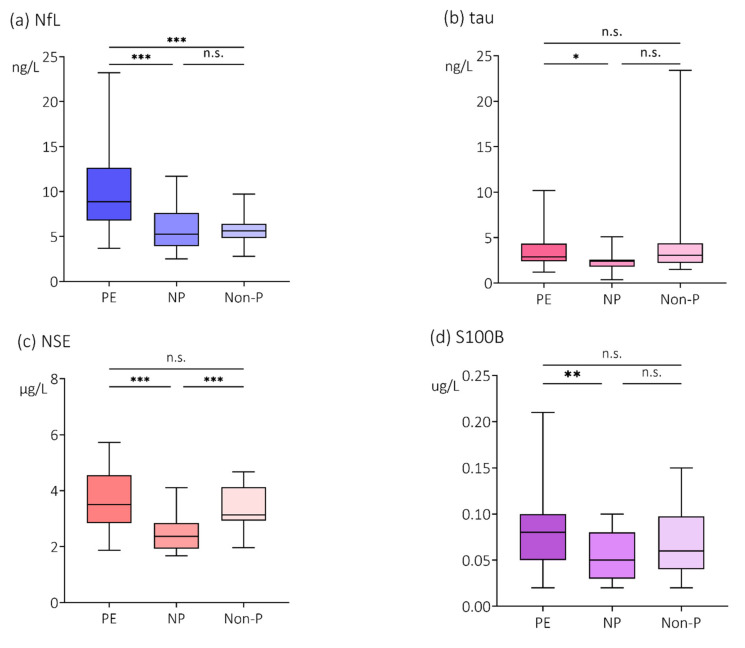
Plasma concentrations of (**a**) NfL, (**b**) tau, (**c**) NSE and (**d**) S100B. Plasma concentrations of the cerebral biomarkers (**a**) neurofilament light (NfL), (**b**) tau, (**c**) neuron specific enolase (NSE) and (**d**) S100B in women with preeclampsia (PE), women with normal pregnancies (NP) and non-pregnant women (Non-P). Values are represented by medians with interquartile range (IQR). Pairwise comparisons by Mann-Whitney U-test, Bonferroni correction. * *p* < 0.05; ** *p* < 0.01; *** *p* < 0.001; n.s. = non-significant.

**Figure 2 cells-11-00789-f002:**
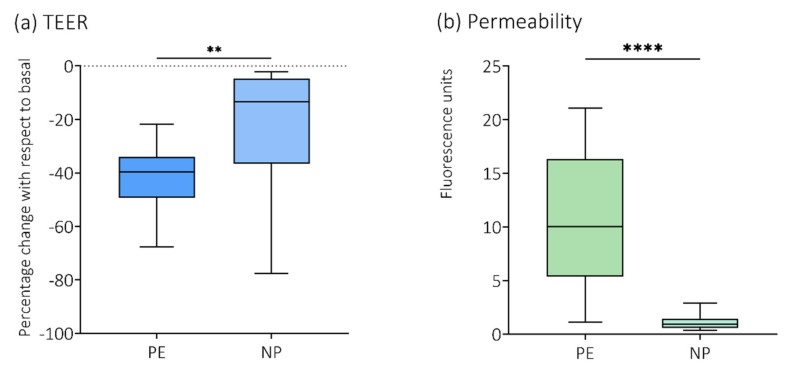
Re−confirmation analysis of TEER and permeability to 70 kDa Dextran. Plasma from a small, randomly chosen sample of the women with preeclampsia (PE, n = 12) and the women with normal pregnancies (NP, n = 13) was analyzed for re−confirmation of previously published data on (**a**) transendothelial electrical resistance (TEER) and (**b**) permeability to 70 kDa fluorescein isothiocyanate (FITC)−dextran [30]. ** *p* < 0.01; **** *p* < 0.0001.

**Figure 3 cells-11-00789-f003:**
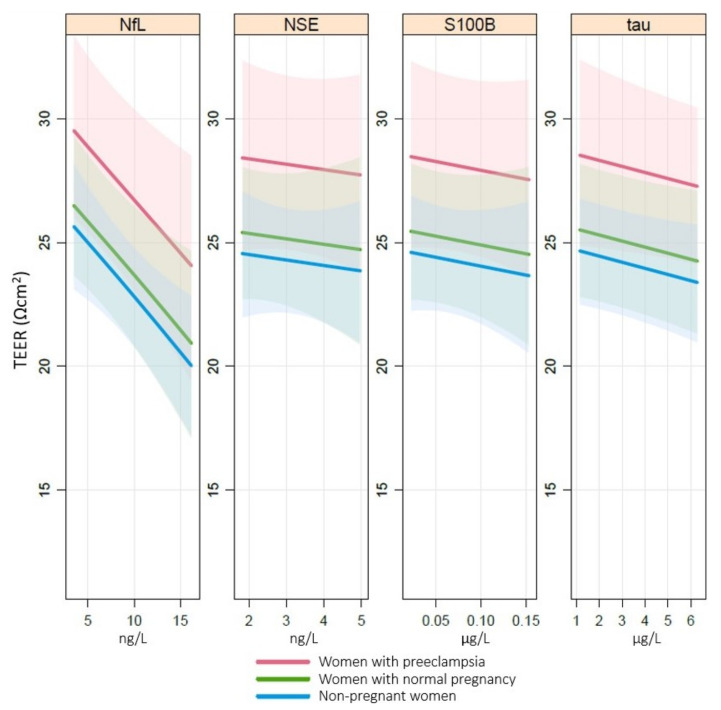
Association of TEER with NfL, tau, NSE and S100B. Associations between transendothelial electrical resistance (TEER (Ωcm^2^)) and cerebral biomarkers neurofilament light (NfL (ng/L)), tau (ng/L), neuron-specific enolase (NSE (µg/L)) and S100B (µg/L) in plasma analyzed by a cumulative probability model stratified by group and adjusted for baseline TEER, body mass index, parity and maternal age. NfL *p* < 0.01, tau *p* = n.s., NSE *p* = n.s., S100B *p* = n.s.

**Table 1 cells-11-00789-t001:** Clinical characteristics of the study population.

Clinical Characteristics	Preeclampsia (n = 28)	Normal Pregnancy (n = 28)	Non-Pregnant (n = 16)	*p*-Values
Maternal age (years)	28 (25–32)	33 (29–35)	27 (24–36)	n.s.
Nulliparous	23 (82%)	10 (36%)	9 (56%)	<0.001
BMI	26 (23–29)	24 (22–26)	22 (20–25)	<0.001
At inclusion				
Gestational week	35 (29–37)	35 (27–38)		n.s.
Blood pressure (mmHg)				
Systolic	150 (140–160)	110 (110–120)	110 (110–118)	<0.001
Diastolic	98 (86–100)	70 (60–75)	70 (65–70)	<0.001
MAP	113 (107–120)	83 (77–90)	83 (80–86)	<0.001
Neurological symptoms (yes)				
Headache	18 (64%)	4 (14%)	3 (19%)	<0.001
Severe headache (VAS ≥ 5)	10 (36%)	0	0	<0.001
Visual disturbances	10 (36%)	0	0	<0.001
Headache & visual disturbances	10 (36%)	0	0	<0.001
Any neurological symptom	20 (71%)	4 (14%)	3 (19%)	<0.001
TEER (Ωcm^2^)				
Baseline value	34.9(29.7–38.8)	33.7 (29.7–42.6)	29.2 (25.4–38.8)	n.s.
After plasma exposure	22.9 (18.1–27.5)	27.1 (18.8–35.5)	23.8 (21.6–29.3)	n.s.
Δ-TEER	11.9 (8.5–14.8)	7.6 (3.7–11.9)	5.8 (2.0–8.0)	<0.001

Data are presented as medians (IQR) or numbers (%). Abbreviations: BMI, body mass index; MAP, mean arterial pressure; VAS, visual analogue scale; TEER, transendothelial electrical resistance; Δ-TEER, the difference between TEER values before and after exposure to plasma; n.s., non-significant.

**Table 2 cells-11-00789-t002:** Characteristics of the women with preeclampsia.

Clinical Characteristics	Preeclampsia (n = 28)
Gestational week at preeclampsia diagnosis	35 (22–41)
Severe preeclampsia at inclusion, n	10 (36%)
Blood pressure medication at inclusion, n	22 (79%)
Magnesium treatment, n	0 (0%)
Gestational week at delivery	35 (25–41)
Severe preeclampsia at delivery, n	16 (57%)

Numbers are presented as median (range) or numbers (%). Severe preeclampsia is defined according to the guidelines from the International Society for the study of Hypertension in Pregnancy (ISSHP).

## Data Availability

The data presented in this study will be anonymized and made available on request from the corresponding author, after approval from the Uppsala Ethical Review Board. The data are not publicly available due to secrecy.

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
