# Peer review of "Cerebral Biomarkers and Blood-Brain Barrier Integrity in Preeclampsia"

_cells, 2022, doi:10.3390/cells11050789_

Round 1

Reviewer 1 Report

I have no comments on the manuscript.

Author Response

We thank the reviewer for taking the time to read through the manuscript once more, we greatly appreciate it!

Reviewer 2 Report

The authors resubmitted the original paper with some improvements regarding the materials and methods, some lacking parameters in the population studied, but none real improvement concerning the BBB properties. Vigh et al. 2021 clearly indicated how the TEER measurements are experimentation and system-dependent and can't be set alone to define the BBB integrity. Moreover, the TEER measurements testify the ion fluxes, and even increased TEER values, the permeability for integrity markers such as sodium fluorescein of Dextrans is not proportionally alter and can be as low as the control conditions. Once again, the relationship between preeclampsia and BBB integrity is quite original, but poorly demonstrated. 

Round 2

Reviewer 2 Report

Thanks to the authors for this serious revision and the data provides as sub-figures or supplementary figures. 

Even for a pilot study, some standards have to be checked to fit in the BBB field. In the proposed revised version, this preliminary study deserves to be published in Cells.

This manuscript is a resubmission of an earlier submission. The following is a list of the peer review reports and author responses from that submission.

Round 1

Reviewer 1 Report

In the manuscript entitled "Cerebral Biomarkers and Blood-Brain Barrier Integrity in Preeclampsia" the authors reported the application of Transendothelial electrical resistance (TEER) to assess blood-brain barrier (BBB) integrity, furthermore, the authors reported a panel of biomarkers used to discriminate preeclamptic versus normal pregnancy women identifying the biomarker neurofilament light chain as a novel marker useful to discriminate PE by linking it to altered BBB permeability. The article is well written and I do not find substantial changes to suggest. 

Reviewer 2 Report

The presented work is exciting and valuable in the context of understanding the pathomechanism of brain disorders in the course of preeclampsia, which has not yet been explained.
After reviewing the manuscript, comments are as follows:
1. In the abstract, it is worth including short information about the studied brain biomarkers.
2. The concept of separating a group of non-pregnant women is not entirely clear – it requires justification.
3. The authors should provide full criteria for diagnosing preeclampsia according to ISHHP, noting that it is not only hypertension with proteinuria, and preeclampsia can also be diagnosed in the absence of proteinuria. The criteria for severe preeclampsia, included in the "Results" chapter (L 180 – 181), should be placed in the “Material and Methods” chapter, with detailed definitions.
4. The term "early pregnancy" needs to be defined (Table 1). Is this the first trimester, pregnancy until the end of 8, 10 or 12 weeks? (L 186).
5. Conclusions should only be drawn from the research presented in the paper. There should be no reference to the results of work previously published (L 346).
6. The manuscript requires minor linguistic corrections: grammatical, but above all stylistic: some sentences are too long and difficult to understand.

Reviewer 3 Report

Despite the major interest suggested by the title, and the potential novelty arising from a relationship between preeclampsia and blood-brain barrier disorders, the overall meaning of a study basing its demonstration only on TEER measurements to emphasize to a potential BBB defects is scientifically limited. I should note that using an in vitro BBB model to point out the main results is correct, but the authors have to use and much more demonstrate how the BBB is altered/touched after treatment, and to submit this paper once done. The conclusions are unfortunately too strong regarding the data proposed.

Some key arguments to help the authors to revise and consolidate it:

  • a better knowledge of the BBB main features would help you to design your appproach and fix the mais key points such as permeability experiments for Dextran or other small integrity markers such as Lucifer yellow of sodium fluorescein. For reviews, Saint-Pol et al., 2020, Cells; Sweaney et al, 2019. TEER measurements reflects the permeability for only ions and is an indicator but not a proof of BBB disruption or Pe modulations. Please refer to Vigh et al., 2021 for more precisions, this review in Micromachine pointed out all the drawbacks of TEER measurement and how to use if as well as possible. Pe and TEER have to be shown in the same experimental set to be sure of the BBB integrity and seems to be possible for the authors since they published Dextran data earlier.
  • IF for tight junction proteins and TJ-associated proteins such as ZO1 would be appreciated, as well as a look at the actin cytoskeleton since its main role in the stabilization of junctions complexes. 
  • Even slight modulation of TEER and P-e I guess, efflux pumps activity can be modified and drive a BBB defect. Pump-out experiments can be done from hCMEC/D3 cells and detailed in Sevin et al., 2019, IJMS.

once again, the novelty of the relationship between preeclampsia and BBB disorders is high, but deserves to be deepened studied.